# SOM2IMG: Self-Organizing Map Based Tabular-to-Image Transformation for Deep Learning

## Abstract

Tabular-to-image (T2I) transformation methods convert structured data into image representations to leverage the power of convolutional architectures on tabular datasets. We present SOM2IMG, a novel Self-Organizing Map (SOM) based approach that transforms tabular data into spatial image representations, facilitating the effective application of Convolutional Neural Networks (CNNs) to tabular datasets. SOM2IMG flips rows and columns so that each feature becomes a point to be organized. The SOM then creates a 2D map that reveals similar features, which serves as a blueprint for arranging features spatially in the generated image. Here, we evaluated SOM2IMG on nine diverse datasets under two comparative setups. First, against seven existing T2I methods paired with a CNN, our SOM2IMG + CNN consistently achieved the top performance across all datasets. Second, compared with other machine learning algorithms (non-neural network) applied directly to tabular data, SOM2IMG + CNN ranked first on 5 datasets and placed 2nd to 3rd on the others, primarily behind boosting algorithms. Statistical analysis against a CNN without T2I (baseline CNN) confirms SOM2IMG's substantial improvement with a large effect size, making SOM2IMG the only method to consistently outperform this baseline CNN. Although boosting methods outperformed SOM2IMG + CNN on some datasets, our results establish a strong benchmark for practitioners seeking to apply CNNs to tabular data. Unlike other transformation approaches, SOM2IMG produces interpretable spatial arrangements where related features cluster together, enabling CNN kernels to capture local feature relationships more effectively while preserving original data values, yielding both improved predictive performance and enhanced interpretability. To our knowledge, SOM2IMG is the first SOM-based T2I transformation method, demonstrating that principled spatial organization potentially offers a genuine advantages for applying deep learning to structured data.

## 1 Introduction

Convolutional Neural Networks (CNNs) have achieved remarkable success in domains with inherent spatial structure, such as computer vision and speech processing, by exploiting local patterns and hierarchical feature representations (Yamashita et al., 2018; Alzubaidi et al., 2021). Their effectiveness stems from architectural inductive biases that assume spatially adjacent features share meaningful relationships. This assumption allows convolutional kernels to capture coherent patterns such as edges, textures, and higher-order visual structures (Lecun et al., 1998; Krizhevsky et al., 2017). However, this spatial assumption becomes problematic when applying CNNs to tabular data, where features typically lack inherent spatial relationships. In tabular datasets, prevalent across healthcare, finance, and scientific domains, features are often independently measured variables with arbitrary column ordering. Without meaningful spatial locality, CNN kernels sweep across unrelated features, reducing their ability to learn coherent representations (Borisov et al., 2024; Grinsztajn et al., 2022). Consequently, some advanced machine learning methods such as gradient boosting and random forests have consistently outperformed CNNs on structured tabular data.

To bridge this gap, Tabular-to-Image (T2I) transformation methods have emerged as a promising approach. These techniques convert tabular feature vectors into image-like spatial representations, en-

abling CNN architectures to process structured data. Existing methods such as DeepInsight (Sharma et al., 2019) and IGTD (Zhu et al., 2021) employ various strategies, from dimensionality reduction to correlation-based clustering, to arrange related features in spatial proximity. However, these approaches often rely on supervised information or lack principled methods for discovering the inherent structure within feature relationships.

In this work, we introduce SOM2IMG, a Self-Organizing Map (SOM) based approach that enables a fundamentally different representational learning strategy for tabular-to-image transformation. Instead of treating samples as data points, SOM2IMG trains on the transpose of the input matrix, where each feature is represented by its values across all samples. This allows the SOM to learn natural clusters and relationships among features in an unsupervised manner, with its 2D grid topology providing a spatial template for image generation. Each feature is mapped to its Best Matching Unit (BMU) coordinates without altering its values, thereby preserving data integrity while enhancing interpretability through spatial clustering of related features.

The central innovation of SOM2IMG lies in its use of SOM-driven representational learning for spatial organization. Unlike prior methods that rely on correlation-based clustering, dimensionality reduction, or brute-force optimization, SOM2IMG leverages the self-organizing properties of SOMs to position features with similar patterns in close proximity. This yields a representation that simultaneously improves predictive performance and offers direct visual insight into feature relationships, all while maintaining the original feature values.

Our key contributions include: (1) the introduction of the first SOM-based tabular-to-image transformation method that creates principled spatial arrangements through unsupervised feature relationship learning; (2) a systematic evaluation points to the method's potential within the T2I CNN paradigm, achieving a perfect win rates (9/9 datasets) against the random (T2I) baseline with an effect sizes of Cohen's $d = +1.08$, pointing to a genuine improvement rather than measurement noise; (3) evidence that our clustering-based spatial strategy consistently outperforms optimization-based T2I techniques, proving that structured spatial organization is non-trivial; and (4) an establishment of a benchmark for practitioners seeking to apply deep learning to tabular data, demonstrating that when CNNs are the desired architecture, SOM2IMG transformation provides the optimal strategy with interpretable feature clustering and superior predictive performance.

## 2 RELATED WORKS

### 2.1 SOM

Self-Organizing Maps (SOMs) have long been used in unsupervised learning and data visualization (Kohonen, 2001). They are especially common for exploring feature relationships and reducing dimensionality so that high-dimensional data can be represented in two dimensions. What has received less attention, however, is the use of SOMs for converting tabular data into image form. Classic SOMs are usually applied to data samples, not features, which is a key distinction from the feature-centered setup required for effective tabular-to-image (T2I) transformation.

SOMs are appealing because they preserve local neighborhoods, provide interpretable 2D maps, and naturally group similar features close together. These properties make them a good fit for T2I tasks, where spatial coherence matters. In contrast, dimensionality reduction methods often distort local neighborhoods, and optimization-based approaches can get stuck in suboptimal layouts. SOMs, by comparison, offer a principled and repeatable way to create spatial organizations that capture both local and global structure.

Several studies have combined SOMs with CNNs in other contexts. For example, Agboka et al. (2025) used a SOM before CNN classification in modeling malaria vector resistance, while Ramirez-Quintana & Chacon-Murguia (2015) applied SOMs for object detection tasks. In these works, though, the SOM was used mainly for clustering or dimensionality reduction in the sample space, not for arranging features into a grid for CNN input. Our work takes a different angle: we apply SOMs directly in feature space, building structured layouts that make tabular data more suitable for CNNs. To our knowledge, this is the first attempt to use SOMs in this way for T2I, producing interpretable, spatially meaningful feature maps.

## 2.2 Existing T2I Algorithms

A number of T2I algorithms have been developed to make CNNs usable on structured data, each using different strategies for arranging features as pixels. Below we summarize the main approaches that we compare against.

**DeepInsight (T2I)** computes feature similarities, projects them into 2D using methods such as t-SNE or kernel PCA, and assigns features to nearby pixels based on this projection. Feature values become pixel intensities, producing grayscale images suitable for CNNs (Sharma et al., 2019).

**IGTD (T2I)** formulates pixel placement as an optimization problem that preserves rank-order distances between features. LM-IGTD extends this by adding a Laplacian-based objective to better handle mixed data types (Zhu et al., 2021).

**REFINED (T2I)** begins with multidimensional scaling, then iteratively adjusts the feature layout with a hill-climbing procedure so that correlated features become neighbors (Bazgir et al., 2021).

**NCTD (T2I)** (Novel Convolutional Transformation for Data) expands the feature space with polynomial interactions, then applies affine transformations to arrange them into RGB images. Each channel stores a different normalized projection of the sample, aiming to capture variability that CNNs typically exploit in natural images (Alenizy & Berri, 2025).

**SuperTML (T2I)** embeds each feature value as a small glyph placed at a fixed coordinate from a lookup table. The resulting "canvas" can be processed with pre-trained CNNs (Sun et al., 2019).

**Correlation-Based Pixel Mapping (CorrelationBased (T2I))** uses correlation or mutual information to place strongly related features near each other and weakly related features farther apart. Despite its simplicity, it often performs competitively with more complex algorithms (Medeiros Neto et al., 2023).

**Random Stack Mapping (Random (T2I))** assigns features to random pixel locations as a baseline. Since no data structure informs the layout, this method highlights how much benefit other approaches provide over random assignment.

Together, these methods cover dimensionality reduction, optimization, correlation-based heuristics, and random baselines. Each has strengths but also clear limitations, such as limited interpretability, reliance on supervised information, or heavy computational cost. These challenges motivate our SOM-based approach, which is designed to provide interpretable, structure-preserving layouts while remaining effective for CNNs.

## 3 Methods and Materials

### 3.1 Datasets

Table 1: Dataset characteristics and model performance summary. N = sample size; Type: Continuous = continuous features only, Mixed = combination of categorical and continuous features, Discrete = discrete categorical features; CNN Competitive = whether CNN-based methods achieved competitive performance compared to traditional ML methods.

| Dataset | N | Features | Type | CNN Comp. | Best Model | F1 |
|---|---|---|---|---|---|---|
| Breast Cancer Wisconsin | 569 | 30 | Cont. | Yes | SOM2IMG (T2I) | 0.96 |
| Heart Disease | 303 | 20 | Mixed | Yes | SOM2IMG/CorrBased/DeepInsight | 0.84 |
| Lung Cancer | 101 | 32 | Discrete | No | Gradient Boosting | 0.54 |
| Myocardial Infarction | 1700 | 184 | Mixed | No | XGBoost | 0.87 |
| Obesity Levels | 2111 | 23 | Mixed | No | XGBoost | 0.96 |
| Heart and Soul | 515 | 100 | Cont. | Yes | SOM2IMG (T2I) | 0.81 |
| Dry Beans | 13612 | 16 | Cont. | No | XGBoost | 0.93 |
| Spambase | 4601 | 16 | Cont. | Yes | SOM2IMG (T2I) | 0.93 |
| Dengue & Chikungunya | 17172 | 26 | Mixed | Yes | SOM2IMG (T2I)/Gradient Boosting | 0.61 |

**Benchmarks from the UCI Machine Learning Repository**   We include seven classic tabular datasets for classification tasks spanning medical diagnostics, agricultural morphology, and text-based spam detection: Breast Cancer Wisconsin (Diagnostic), Heart Disease (Cleveland subset),

Lung Cancer, Myocardial Infarction (Complications), Obesity Levels, Dry Beans, and Spambase. These datasets feature a mix of continuous and categorical inputs and serve as widely-adopted baselines in structured data research (Kelly et al., 2025).

**Proprietary Heart and Soul Study** The Heart and Soul dataset was contributed by our collaborators. It derives from a prospective cohort (2000–2002) of patients with stable coronary heart disease in Northern California, originally enrolling 1,024 participants at Veterans Affairs Medical Centers, a university hospital, and community clinics. As shown in Table 1, our subset includes 515 individuals with roughly 100 clinical, laboratory, and psychometric variables, among them PHQ-9 depression scores ($\geq$10 indicating depression), inflammatory biomarkers, and standard labs enabling predictive modeling of depression and other outcomes in patients.

**Arboviral Cases from Mendeley Data** We also used a real-world epidemiological dataset of Dengue and Chikungunya cases from Recife, Brazil (2015–2020). This collection comprises over 17,000 cleaned records of confirmed and ruled-out arbovirus infections, with clinical, sociodemographic, and laboratory features, providing a challenging multi-class public-health classification task (Tabosa et al., 2021).

## 3.2 SOM2IMG: TRANSFORMATION

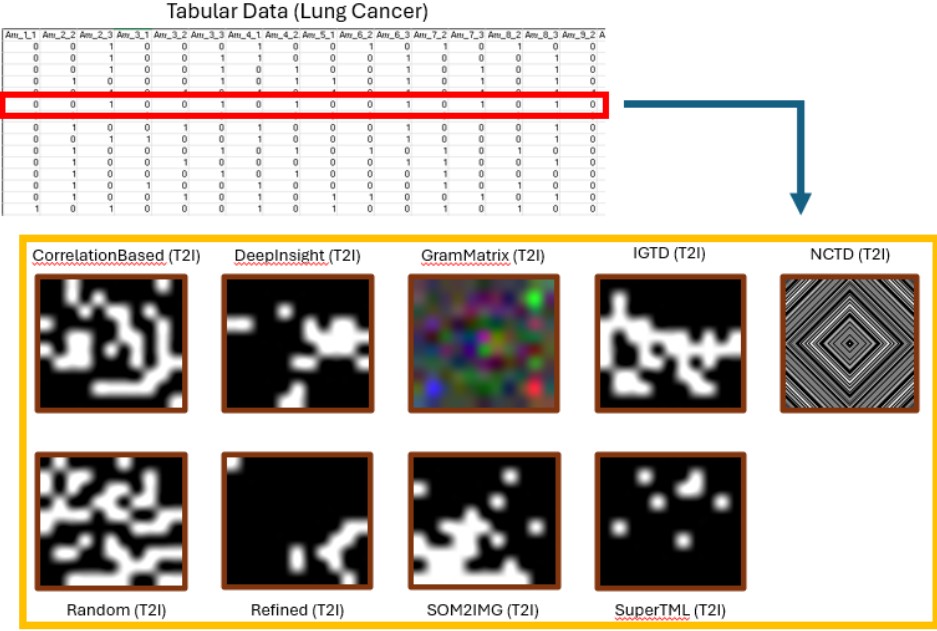

Figure 1: Tabular-to-image (T2I) transformation methods applied to lung cancer data. Top panel shows a sample of the original tabular data matrix with binary features. Bottom panel displays the resulting image representations from the SOM2IMG, and Random (T2I) stacking baseline, as well as seven different T2I methods: CorrelationBased, DeepInsight, GramMatrix, IGTD, NCTD, Random, Refined, SOM2IMG, and SuperTML. Each method generates distinct visual patterns that encode the tabular relationships differently.

Our method, SOM2IMG, takes a new angle on converting tabular data into images by using a Self-Organizing Map (SOM) to arrange features on a 2D grid. Instead of clustering samples, as SOMs normally do, we train the SOM on the features themselves. This way, features that behave similarly across the dataset are placed near each other on the grid. Each sample is then turned into an image by placing its feature values at the grid locations assigned to their corresponding features. The end result is an image where related features appear close together, creating a spatial structure that CNNs can process effectively. A sample of the converted data can be seen in Figure 1.

The process has three steps:

**Phase 1: Representing Features.** For a dataset $X \in \mathbb{R}^{n \times d}$ with $n$ samples and $d$ features, we transpose it to $X^T \in \mathbb{R}^{d \times n}$. Now, each feature is represented as a vector of its values across all samples. This flips the focus from samples to features.

**Phase 2: Training the Self-Organizing Map.** We set up a 2D SOM grid with dimensions $h \times w$, calculated as:

$$h = \lceil \sqrt{d} \rceil, \quad w = \lceil \frac{d}{h} \rceil \tag{1}$$

This creates a square-like grid that fits all $d$ features.

During training, each feature finds its Best Matching Unit (BMU):

$$\text{BMU}(\mathbf{f}_i) = \arg \min_j ||\mathbf{f}_i - \mathbf{w}_j||_2 \tag{2}$$

where $\mathbf{f}_i$ is the $i$-th feature vector across all samples, and $\mathbf{w}_j$ is the weight vector of grid neuron $j$. Similar features end up close together on the grid, creating a topology-preserving feature map.

**Phase 3: Generating Images.** Once training is complete, each grid position corresponds to a specific feature. For any new sample, we create an image by placing each feature's value at the pixel location given by its BMU. Empty spots are filled with zeros. The result is a structured image that preserves the raw feature values but arranges them in a way that reflects their learned relationships.

This method keeps the original tabular information intact while reordering it into a spatial layout that CNNs can effectively process.

### 3.3 MACHINE LEARNING METHODS

To establish strong baselines, we evaluated each dataset using four broad classes of well-understood machine learning (ML) classifiers on the tabular data (without transformation), with default hyper-parameters and minimal tuning.

**Tree-based ensembles** These methods aggregate many decision trees to reduce variance and improve generalization on structured data. We included Random Forest (Breiman, 2001), and two boosting ensembles, Gradient Boosting (Friedman, 2001) and XGBoost (Chen & Guestrin, 2016), which together set the performance ceiling across our nine tabular datasets.

**Kernel and linear methods** For interpretable, margin-based and probabilistic modeling, we employ an RBF-kernel SVM (Cortes & Vapnik, 1995) and standard Logistic Regression (Cox, 1958). These represent strong mid-range baselines that capture both non-linear boundaries and linear log-odds relationships.

**Instance-based learning** k-Nearest Neighbors (Cover & Hart, 1967) serves as a non-parametric local benchmark, predicting class labels by majority vote among the closest training points under a chosen distance metric.

**Feed-forward neural network** A three-layer ReLU MLP (Rumelhart et al., 1986) (100–100–100 units) provides a generic non-convex function approximator. It was trained with early stopping on a held-out validation split, offering insight into the value of generic neural architectures versus specialized T2I representations.

### 3.4 CNN ARCHITECTURE

To evaluate the images generated by the T2I transformation methods, we employed a standardized Convolutional Neural Network (CNN) architecture inspired by the seminal LeNet-5 model (Lecun et al., 1998). This choice provides a well-established and simple baseline to fairly assess the quality of the image representations themselves, rather than the power of an overly complex deep learning model. The architecture is kept consistent for all experiments.

The model processes an input image through two sequential blocks, each containing a convolutional layer followed by a max-pooling layer. The first convolutional layer uses 16 filters of size 5×5, and

the second uses 32 filters of the same size, both with ReLU activation. Each convolutional layer is followed by a 2×2 max-pooling layer for dimensionality reduction. The resulting feature maps are then flattened into a one-dimensional vector and passed to a dense, fully-connected layer with 128 neurons (ReLU activation). The final dense layer produces the class probabilities using a softmax activation function.

### 3.5 PIPELINE

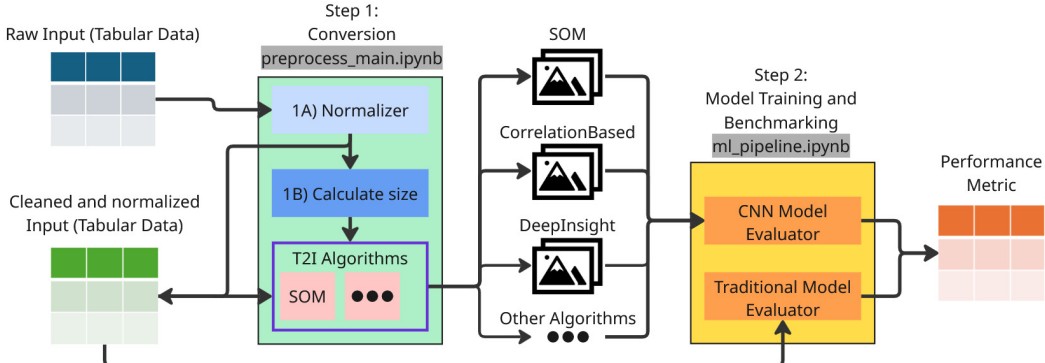

Figure 2: Overview of the two-stage automated pipeline. In Stage 1, raw tabular data are min–max normalized, and transformed by multiple T2I algorithms (e.g., SOM2IMG, CorrelationBased, DeepInsight), with outputs saved as both `.png` and `.json` alongside mapping keys. In Stage 2, seven traditional ML models on the original tabular data and a standardized CNN on the generated images are benchmarked in parallel using 10-fold cross-validation, yielding a direct comparison of performance across methods.

To ensure rigorous and reproducible comparison, we designed a two-stage automated pipeline, as illustrated in Figure 2. Stage 1 converts tabular datasets into image representations using multiple T2I algorithms; Stage 2 benchmarks both traditional ML models on original data and CNNs on generated images.

#### 3.5.1 DATA PREPROCESSING AND IMAGE GENERATION

For each dataset, we apply consistent preprocessing: (1) Min-Max normalization to [0,1] range to ensure scale-invariant comparisons across T2I methods and distance-based algorithms, (2) optimal rectangular image dimension calculation to create near-square aspect ratios while maintaining consistent dimensions across T2I algorithms within each dataset, and (3) systematic application of all nine T2I algorithms to generate corresponding images for every sample. Generated images are stored as both `.png` files for visualization and `.json` arrays for CNN input, with `key.csv` files maintaining sample-to-label mappings.

#### 3.5.2 MODEL TRAINING AND BENCHMARKING

The evaluation stage proceeds along two parallel tracks: various ML models train on original normalized tabular data while CNNs train on T2I-generated images. All models use 10-fold cross-validation for robust performance estimation, with fallback to 10 independent random trials for datasets with insufficient minority class samples for stratified splitting.

## 4 RESULTS

SOM2IMG achieves the best performance on three datasets and ties on two datasets, for a total of five out of nine datasets. Notably, SOM2IMG dominates on datasets with continuous features (Breast Cancer Wisconsin, Heart and Soul, Spambase) and performs well on mixed feature types (Dengue & Chikungunya). The method shows particular strength on smaller to medium-sized feature sets (16-100 features). The full F1 score result can be seen in Appendix A Table 3.

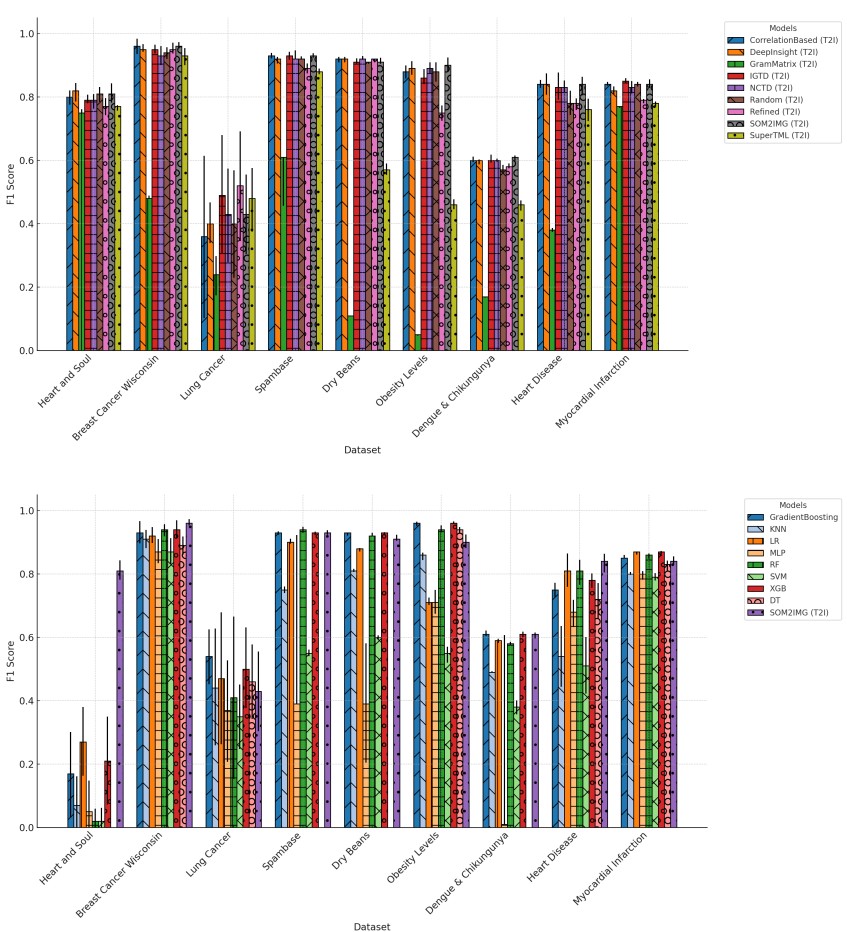

Figure 3: Comparison of F1 scores (bars) with 95 % confidence intervals (error bars) across nine benchmark datasets. **Top panel:** Tabular-to-Image (T2I) transformation methods, including *SOM2IMG*. **Bottom panel:** Conventional tabular-learning baselines plus *SOM2IMG*.

When comparing against only other T2I algorithms, SOM2IMG demonstrates competitive performance across eight out of nine datasets, consistently achieving first place or tying for the top position (Figure 3, top panel). When compared against all machine learning methods, SOM2IMG remains competitive on five out of nine datasets, maintaining its position among the best-performing approaches (Figure 3, bottom panel).

On datasets where CNNs are competitive with traditional methods (5 out of 9 datasets), SOM2IMG is always the top performer or ties. For datasets where traditional methods dominate (Lung Cancer, Myocardial Infarction, Obesity Levels, Dry Beans), XGBoost and Gradient Boosting maintain their advantage, suggesting that T2I transformation are conditionally beneficial depending on a dataset's characteristics.

## 4.1 SOM2IMG STATISTICAL ANALYSIS

We conducted statistical comparisons between SOM2IMG and other T2I methods against a random pixel arrangement baseline (Random T2I). For each method, we calculated the mean F1 score difference (Avg $\Delta$) relative to Random (T2I) arrangement across all datasets, counted wins (datasets where the method outperformed random), and computed *Cohen's d effect size* to quantify the magnitude of performance differences. Cohen's d normalizes the mean difference by the pooled standard deviation, providing a standardized measure of practical significance independent of dataset scale. Table 2 presents the results.

**Cohen's d Effect Size:** To quantify the practical significance of performance differences independent of dataset scale, we computed Cohen's d:

$$d_M = \frac{\bar{X}_M - \bar{X}_{\text{random}}}{s_{\text{pooled}}} \tag{3}$$

where $\bar{X}_M$ and $\bar{X}_{\text{random}}$ are the mean F1 scores across all datasets for method $M$ and the random baseline, respectively, and $s_{\text{pooled}}$ is the pooled standard deviation:

$$s_{\text{pooled}} = \sqrt{\frac{(n_M - 1)s_M^2 + (n_{\text{random}} - 1)s_{\text{random}}^2}{n_M + n_{\text{random}} - 2}} \tag{4}$$

where $n_M$ and $n_{\text{random}}$ are the sample sizes, and $s_M$ and $s_{\text{random}}$ are the standard deviations for method $M$ and the random baseline, respectively. This provides a standardized measure of practical significance with established interpretations: $|d| \geq 0.8$ indicates a large effect, $0.5 \leq |d| < 0.8$ indicates a medium effect, and $|d| < 0.5$ indicates a small effect.

### 4.1.1 STATISTICAL METRICS AND FORMULATION

For each method $M$ and dataset $D$, we computed three key statistical measures:

**Mean F1 Score Difference (Avg $\Delta$):** The average improvement in F1 score relative to the random baseline across all datasets:

$$\text{Avg } \Delta_M = \frac{1}{|D|} \sum_{d \in D} (F1_{M,d} - F1_{\text{random},d}) \tag{5}$$

where $F1_{M,d}$ represents the F1 score of method $M$ on dataset $d$, and $F1_{\text{random},d}$ is the F1 score of the random baseline on the same dataset.

**Win Count:** The number of datasets where method $M$ outperforms the random baseline:

$$\text{Wins}_M = \sum_{d \in D} \mathbb{I}(F1_{M,d} > F1_{\text{random},d}) \tag{6}$$

where $\mathbb{I}(\cdot)$ is the indicator function.

Table 2: Statistical comparison of T2I methods against random baseline. Avg $\Delta$ shows mean F1 improvement over random arrangement; Effect Size (Cohen's $d$) indicates practical significance: $|d| \geq 0.8$ (large), $0.5 \leq |d| < 0.8$ (medium), $|d| < 0.5$ (small). As highlighted, SOM2IMG's performance is statistically better than others.

| Model | Avg $\Delta$ | Wins (out of 9) | Effect Size |
|---|---|---|---|
| **SOM2IMG** (T2I) | **+0.021** | **9/9** | **+1.08** |
| IGTD (T2I) | +0.021 | 7/9 | +0.79 |
| CorrelationBased (T2I) | +0.009 | 6/9 | +0.91 |
| NCTD (T2I) | +0.010 | 5/9 | +0.77 |
| DeepInsight (T2I) | +0.012 | 7/9 | +0.55 |
| REFINED (T2I) | -0.012 | 4/9 | -1.30 |
| SuperTML (T2I) | -0.108 | 1/9 | -7.20 |

SOM2IMG demonstrates the most consistent performance among T2I methods, improving over the random baseline on all nine datasets (9/9 wins) with an effect size (Cohen's d = +1.08). This consistency distinguishes SOM2IMG from other methods that show more variable performance across different datasets.

## 5 DISCUSSION AND FUTURE WORK

### 5.1 SOM2IMG PERFORMANCE ANALYSIS

Experimental results suggest that SOM2IMG performs well among T2I transformation methods for CNN applications on these datasets. When comparing against other T2I algorithms, SOM2IMG outperforms other T2I algorithms on 8 out of 9 datasets (Figure 3).

The consistent improvement over random arrangement indicates that SOM2IMG creates meaningful spatial structure. Other methods such as IGTD and DeepInsight show improvements on 7/9 datasets but lack SOM2IMG's consistency across all datasets. Some established methods (REFINED, SuperTML) consistently underperform random arrangement, suggesting that not all spatial transformations benefit CNN learning.

The SOM approach's effectiveness likely stems from its unsupervised learning of feature relationships. By treating features as data points characterized by their value distributions, SOM creates spatial arrangements where similar features cluster together. This provides CNNs with local patterns that may be useful for classification.

### 5.2 COMPARISON WITH OTHER ML MODELS

Some advanced methods such as XGBoost remain competitive overall, achieving top scores on 4 datasets with faster training times (under 0.01 minutes vs. 4.5 minutes for CNN pipeline). The additional preprocessing for tabular-to-image conversion nearly doubles total runtime.

However, this work addresses a specific question: when CNNs are desired for tabular data—whether for transfer learning, integration with deep learning pipelines, or leveraging spatial processing—which transformation method works best? Our results indicate SOM2IMG is an effective choice for these scenarios.

SOM2IMG offers additional benefits beyond accuracy. It preserves original feature values while creating interpretable spatial arrangements where related features cluster together. This provides both predictive performance and visual insight into feature relationships, unlike traditional methods that offer limited interpretability.

### 5.3 LIMITATIONS AND FUTURE DIRECTIONS

This study evaluates nine datasets, some relatively small or domain-specific. Larger benchmarks spanning different data characteristics would strengthen validation of the approach.

Future work should examine modern CNN architectures beyond the LeNet-inspired baseline and explore hybrid approaches combining SOM2IMG transformation with traditional preprocessing. The success of treating features as data points suggests broader applications beyond CNN transformation, potentially informing feature engineering and interpretability methods.

Cost-benefit analyses incorporating deployment requirements will guide practical adoption. While traditional methods offer computational efficiency for general classification, scenarios requiring CNN integration or spatial feature visualization may justify the transformation overhead.

### LLM USAGE DECLARATION

The authors acknowledge the use of Claude (Anthropic) and ChatGPT (OpenAI) to revise portions of this manuscript, some code debugging, and to locate relevant previous work during initial research.

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

## A  APPENDIX

Table 3: F1 scores (rounded to two decimal places) across 17 models and 9 datasets. The competitive models are bolded. Traditional methods (top section) dominate on 4 datasets, while tabular-to-image methods with CNNs (bottom section) excel on 5 datasets. SOM2IMG achieves best or tied performance on 5/9 datasets and is the only T2I method to improve over random baseline on all datasets, pointing to a consist improvement among transformation approaches.

| Model | Breast Cancer Wisconsin | Heart Disease | Lung Cancer | Myocardial Infarction | Obesity Levels | Heart and Soul | Dry Beans | Spambase | Dengue and Chikungunya |
|---|---|---|---|---|---|---|---|---|---|
| DT | 0.89 | 0.72 | 0.46 | 0.83 | 0.94 | 0.34 | 0.89 | 0.88 | 0.54 |
| GradientBoosting | 0.93 | 0.75 | **0.54** | 0.85 | **0.96** | 0.17 | **0.93** | **0.93** | **0.61** |
| KNN | 0.91 | 0.54 | 0.44 | 0.80 | 0.86 | 0.07 | 0.81 | 0.75 | 0.49 |
| LR | 0.92 | 0.81 | 0.47 | **0.87** | 0.71 | 0.27 | 0.88 | 0.90 | 0.59 |
| MLP | 0.87 | 0.68 | 0.37 | 0.80 | 0.71 | 0.05 | 0.39 | 0.39 | 0.01 |
| RF | 0.94 | 0.81 | 0.41 | 0.86 | 0.94 | 0.02 | 0.92 | 0.94 | 0.58 |
| SVM | 0.87 | 0.51 | 0.35 | 0.79 | 0.55 | 0.02 | 0.60 | 0.55 | 0.38 |
| XGB | 0.94 | 0.78 | 0.50 | **0.87** | 0.96 | 0.21 | **0.93** | **0.93** | **0.61** |
| CorrelationBased (T2I) | 0.96 | **0.84** | 0.36 | 0.84 | 0.88 | 0.80 | 0.92 | **0.93** | 0.60 |
| DeepInsight (T2I) | 0.95 | **0.84** | 0.40 | 0.82 | 0.89 | 0.82 | 0.92 | 0.92 | 0.60 |
| GramMatrix (T2I) | 0.48 | 0.38 | 0.24 | 0.77 | 0.05 | 0.75 | 0.11 | 0.61 | 0.17 |
| IGTD (T2I) | 0.95 | 0.83 | 0.49 | 0.85 | 0.86 | 0.79 | 0.91 | **0.93** | 0.60 |
| NCTD (T2I) | 0.93 | 0.83 | 0.43 | 0.83 | 0.89 | 0.79 | 0.92 | 0.92 | 0.60 |
| Random (T2I) | 0.94 | 0.78 | 0.40 | 0.84 | 0.88 | **0.81** | 0.91 | 0.92 | 0.57 |
| REFINED (T2I) | 0.95 | 0.78 | 0.52 | 0.79 | 0.75 | 0.77 | 0.92 | 0.89 | 0.58 |
| **SOM2IMG (T2I)** | **0.96** | **0.84** | 0.43 | 0.84 | 0.90 | 0.81 | 0.91 | **0.93** | **0.61** |
| SuperTML (T2I) | 0.93 | 0.76 | 0.48 | 0.78 | 0.46 | 0.77 | 0.57 | 0.88 | 0.46 |