# OpenReview forum: "SOM2IMG: Self-Organizing Map Based Tabular-to-Image Transformation for Deep Learning"
_ICLR.cc/2026/Conference — Submitted to ICLR 2026_

### Official Review · Reviewer_YjjL · 2025-10-21

**Soundness:** 3
**Presentation:** 3
**Contribution:** 1
**Rating:** 2
**Confidence:** 4

**Summary:**

The authors solve tabular deep learning by converting tables to images and applying conv. nets. The primary novelty is in using sel-organizing maps (SOM) to position the samples in the image in a semantically meaningful way. On a small sample of 9 datasets, the proposed method achieves 5 wins when compared against gradient boosted trees.

**Strengths:**

1. **Clarity.** The paper is clear and well-illustrated. The basics of the method are well explained.
2. **Problem motivation.** The problem of tabular deep learning is an interesting and timely one. There is a plethora of recently proposed methods, and despite a steady progress, there remains a large number of important issues and unanswered questions in the field.

**Weaknesses:**

I was disappointed to see this work being presented in its current form, especially given the significant progress that has been made in the tabular deep learning field. I have several concerns that I would like to highlight:
1. The idea of transforming tabular data into images seems unnecessary and is questionable. The only motivation that I see is that CV field is more mature, so transforming table into images allows us to unlock new methods. While it's true that the CV field has made great advancements, I believe that developing distinct / bespoke methods for tabular data is crucial. Different modalities have unique characteristics, and tailored approaches often lead to better results. For instance, the transformer architecture in NLP and convolutional networks in CV are exemplary models that have emerged from modality-specific research.
2. Even if one accepts the proposed approach, I was underwhelmed by the evaluation. The authors tested their method on a limited sample of 9 datasets, which is a small fraction compared to the hundreds of datasets used in benchmarks like TALENT. Furthermore, the selected datasets are relatively small and performance-saturated, with only two having more than 10k samples. The inclusion of the Wisconsin Breast Cancer dataset (easy to overfit), a toy dataset from the Sklearn library, raises serious concerns about the evaluation's validity as a whole.
3. I was surprised that the authors didn't compare their results to other state-of-the-art methods like TabM, RealMLP, and TabPFN. Including these comparisons would have provided a more comprehensive understanding of the proposed method's strengths and weaknesses.

**Questions:**

Please, see weaknesses. I don't have technical questions per se, except that I totally don't understand how to position this work within the current tabular deep learning literature. Here are the two major questions:
1. What's the upside of transforming tables into images?
2. Why don't the authors compare on larger benchmarks and don't include current SOTA tabular deep learning methods?

---

### Official Review · Reviewer_9bio · 2025-10-27

**Soundness:** 2
**Presentation:** 3
**Contribution:** 1
**Rating:** 2
**Confidence:** 4

**Summary:**

This paper proposes a method for converting tabular data into images to leverage the classification power of CNN.
The idea does make sense, benefiting non-image data by applying CNN.
The paper is generally easy to understand. However, it contains technical unclarity that reduces the quality and mainly the reproducibility of this paper.

**Strengths:**

1. The idea of converting non-image data to images to apply CNN makes sense.
2. The paper is generally well written and easy to understand.

**Weaknesses:**

1. The primary weakness is the lack of explanation on how the proposed SOM2IMG creates an image for a new input.
    On pg. 5, in "Phase 3," the authors wrote, "For any new sample, we create an image by placing each feature's value at...."
    This information alone is not sufficient for reproducing the experiment.
    For example, in this SOM, each node represents one feature, and hence w_j \in R^n (since the original training data set contains n samples), whereas for the new data, the j-th feature f_j \in R^1. To resolve this problem, the authors filled the empty spots with 0. As 0 becomes dominant, the reviewer cannot understand how the authors relate the new input to the existing SOM for creating a new image. The lack of clarity is the primary weakness of this paper.

2. The datasets contain mainly small-sized problems with small numbers of classes. The authors mention this limitation in the Discussion. But the reviewer thinks it is important for the authors to demonstrate the scalability of the proposed method across problems of varying sizes.

3. There are no clear advantages of using the proposed method, as the results in Table 3 do not demonstrate significant superiority with regard to other methods.

4. In the abstract, the authors mention the interpretability of the spatial arrangement. But there is no further explanation on how to leverage the interpretability. For example, how interpretability can generate new insights for human users.

**Questions:**

Suggestions:

1. Please give some examples of images generated from new inputs for the same and contrasting classes, regarding the Lung Cancer data, similar to Fig. 1.

2. Please demonstrate the performance of the proposed method against data with larger number of classes (for example 10) and larger size.

3. The authors mentioned that, to their knowledge, SOM2IMG is the first SOM-based T2I transformation. However, in the past, there was a method for transforming non-image data using SOM and feeding it to a CNN. Furthermore, the past method leverages the visibility of inputs to enhance transparency in the decision-making process.

   K. Ogawa, P. Hartono, Collaborative General Purpose Convolutional Neural Network, Journal of Signal Processing, Vol. 25, No. 2, pp.53-61 (2021) DOI: 10.2299/jsp.25.53

   There was also a neural network that, in its hidden layer, directly converts tabular data into a topological map, similar to SOM, and performs classification.

  P. Hartono, P. Hollensen, T. Trappenberg, Learning-Regulated Context Relevant Topographical Map, IEEE Trans. on Neural Networks and Learning Systems, Vol. 26, No. 10, pp. 2323-2335 (2015). DOI:10.1109/TNNLS.2014.2379275

Please compare the current work with these past works and explain its advantages.

---

### Official Review · Reviewer_VfRd · 2025-10-30

**Soundness:** 1
**Presentation:** 1
**Contribution:** 2
**Rating:** 2
**Confidence:** 4

**Summary:**

This paper proposes SOM2IMG, a framework that converts tabular data into images which can then be used to train CNNs for classification tasks. The authors experiment with 9 datasets and demonstrate that this method outperforms other Table-to-Image (T2I) methods in terms of predictive performance. However, the results also show that standard ML methods still show an overall stronger performance than the proposed method.

**Strengths:**

- I think this paper tackles an interesting task, and the idea has a lot of potential.
- Among T2I methods considered in this work, the proposed method does show improvements.

**Weaknesses:**

- The main result is that this method outperforms against **other T2I** methods. But T2I already isn't a very competetive approach for tabular data, and the results against other ML baselines show that this method is not very competetive in general -- especially since the default params are used for these baselines.
- Overall, I think the current scope of the paper -- using CNNs for tabular classification -- is not well justified (if at all). In fact, I don't see any justification of why this is a problem to begin with. (I do think it is interesting, but I don't see any motivation in the paper).
- The baseline comparison is quite limited. It is hard to say much about the true full performance of a model if it is just using default parameters. I would suggest at least tuning the baseline ML models to get a proper understanding of the performance gap.

**Questions:**

- how are the discrete (categorical) features handled for image generation?
- Is the resulting image also useful for other classifiers? (i.e. what happens when we just flatten the image and use XGBoost/MLP on it? )
- Is it possible to do multi-modal learning with this approach? (e.g. datasets that have images + tabular, these are quite common in medical applications as far as I am aware)
- Figure 1 can potentially be nice. But I'm not sure what I'm supposed to take away from it. Is it just to showcase the diffrence in patterns? Is there something that SOM2IMG produced that other methods are lacking?
- Figure 3 is very hard to read. I suggest either presenting it in a different form (table with bold/italicize?) or at least using larger fonts.
  - Except for the performance of Heart and Soul on the bottom bar chart of Figure 3, I would not use the word _dominates_.
- I think the dataset selection can be better justified as well. Why pick such datasets? (widely used in prev. work? or to add diversity? or because its domain is well-suited? etc.)
- > When compared against all machine learning methods, SOM2IMG remains competitive on five out of nine datasets, maintaining its position among the best-performing approaches
  - Which are the 5? would be nice if that was specified. What is meant by "maintaining among the best" exactly? It would be better to use a more clear comparative metric like rankings to support statements like this.
- What is going on with the "Hearts and Soul" dataset? How come nearly all baselines fail completely and only T2I methods work on it? Any intuition behind why this happens?

Overall, I think the idea of the paper is interesting. However, the current set of results suggests that examining this method _only_ through the lense of tablular classification is not sufficient to justify its usefulness. I would suggest either expanding the scope of the paper (examining multi-modal learning, or other classifiers on the generated images) or a further analysis into _when_ and _why_ this method _can_ be useful (e.g. are there certain characterestics of datasets where SOM2IMG is _competetive with/outperforms_ SotA?)

Small notes:

- The metrics should be introduced before the results are discussed (eq. 3 - 6). It's a bit distracting to jump between metrics and results.

References:

- TabPFN: https://github.com/PriorLabs/TabPFN

---

### Official Review · Reviewer_ktS7 · 2025-11-01

**Soundness:** 3
**Presentation:** 2
**Contribution:** 2
**Rating:** 4
**Confidence:** 4

**Summary:**

This paper proposes SOM2IMG, a new Tabular-to-Image (T2I) transformation methodology for effectively applying CNNs to tabular data. The key idea is to apply a Self-Organizing Map (SOM) not to data samples but to the transpose of the feature matrix, thereby generating a 2D map that, by design, places similar features spatially adjacent. Across an extensive evaluation on nine datasets, SOM2IMG shows significant performance gains over conventional Random T2I on every dataset, and achieves the most consistently top-tier performance among CNN-based T2I approaches, establishing a new benchmark in the T2I field.

**Strengths:**

Principled, Consistent Spatial Organization: Unlike previous dimensionality reduction or optimization-based T2I methods, this approach leverages the unsupervised feature-clustering property of SOM to generate an interpretable spatial layout that preserves relationships among features.


Strongest Statistical Consistency Among T2I Methods: It is the only T2I method that shows performance improvement over Random T2I across all nine datasets, with an effect size of Cohen’s d=+1.08—meeting the “large effect” threshold (∣d∣≥0.8)—thus providing strong statistical support for its practical contribution.


Enhanced Interpretability and Data Preservation: While preserving the original feature values, the method arranges related features to form spatial clusters, thereby improving predictive performance while offering enhanced visual interpretability.

**Weaknesses:**

(Novelty)
The proposed SOM-based T2I methodology remains at the level of a clever refinement or variation, rather than a fundamental paradigm shift from existing T2I studies such as DeepInsight and IGTD. Using SOM for data visualization and clustering is an already well-established approach (Kohonen, 2001), and while the main contribution lies in applying SOM to the feature space rather than the sample space, this can be regarded as a natural adjustment to achieve the T2I objective of feature arrangement.

(Technical Quality)
Limited range of CNN architectures: All T2I experiments were conducted using only a simple LeNet-5–based CNN architecture (Sec. 3.4). There is no additional experimental evidence to determine whether the effectiveness of the proposed method generalizes across modern CNN architectures (e.g., ResNet, ViT) or is restricted to lightweight models.

Severe lack of reproducibility information: The paper provides no details on SOM training parameters (learning rate, neighborhood function/radius, number of epochs) or CNN training hyperparameters (learning rate, batch size) (Sec. 3.2, 3.4). This omission constitutes a critical flaw that prevents reproduction of the core experimental results.

Lack of ablation study: There is no quantitative ablation analysis isolating the contribution of key methodological components—such as using the transposed feature matrix and the zero-filling strategy to handle empty grid spaces (Sec. 3.2)—making it unclear how each element affects performance.

(Significance)
The proposed method (SOM2IMG + CNN) is substantially outperformed by tabular-data SOTA models such as XGBoost and Gradient Boosting on 4 out of 9 datasets (Lung Cancer, Myocardial Infarction, Obesity Levels, and Dry Beans) (Table 1, “CNN Comp.” = No). This indicates insufficient evidence that the work meaningfully advances the field for general tabular-data problems where CNNs are not inherently necessary. The performance gains appear limited to scenarios where CNNs are explicitly required.

(Writing & Presentation)
The omission of essential hyperparameter settings for SOM and CNN training severely hinders readers from fully understanding and validating the proposed method (Sec. 3.2, 3.4).

**Questions:**

Verifying CNN architecture generalization: To demonstrate that the utility of the proposed method is not tied to a specific CNN structure (LeNet-5–based), could you present additional experimental results using modern/strong image-classification architectures such as ResNet-18 or ViT (Vision Transformer)? This will allow us to assess whether the SOM2IMG transformation is truly general-purpose.

Disclosing parameters for reproducibility: For fair comparison and reproducibility of T2I + CNN methods, please clearly provide detailed settings and tuning strategies for the SOM training parameters (learning rate, neighborhood radius, number of epochs) and the CNN training hyperparameters (learning rate, batch size).

Ablation study of key components: To complement the currently missing ablations, could you quantitatively report (1) the effect of using the transposed feature matrix and (2) the impact of the zero-filling strategy used to populate the image grid? This will help determine the core contributions of the SOM2IMG components.

Statistical comparison with SOTA ML models: To show that the proposed method is statistically on par with—or significantly superior to—the tabular SOTA (e.g., XGBoost), could you provide statistical significance test results comparing SOM2IMG + CNN against XGBoost? (At present, Table 2 reports only comparisons with Random T2I.) This will help us judge the practical contribution to tabular classification.

---

### Meta-Review · Area_Chair_FGKe · 2026-01-04

**Summary:**

The paper received an average rating of 2.5. Specifically, all reviewers initially leaned towards rejection in their original reviews (2, 2, 2, 4). The authors did not provide a rebuttal to address the reviewers' valuable questions and concerns. Thus, the AC recommended a rejection.

**Reviewer Concerns:**

No rebuttal is provided.

**Reviewer Scores:**

No rebuttal is provided.

---

### Decision · Program_Chairs · 2026-01-26

Reject